# Molecular mechanism of mRNA repression in *trans* by a ProQ-dependent small RNA

Alexandre Smirnov[1,†] 🆔, Chuan Wang[1,†], Lisa L Drewry[1] & Jörg Vogel[1,2,*] 🆔

## Abstract

Research into post-transcriptional control of mRNAs by small noncoding RNAs (sRNAs) in the model bacteria *Escherichia coli* and *Salmonella enterica* has mainly focused on sRNAs that associate with the RNA chaperone Hfq. However, the recent discovery of the protein ProQ as a common binding partner that stabilizes a distinct large class of structured sRNAs suggests that additional RNA regulons exist in these organisms. The cellular functions and molecular mechanisms of these new ProQ-dependent sRNAs are largely unknown. Here, we report in *Salmonella* Typhimurium the mode-of-action of RaiZ, a ProQ-dependent sRNA that is made from the 3′ end of the mRNA encoding ribosome-inactivating protein RaiA. We show that RaiZ is a base-pairing sRNA that represses in *trans* the mRNA of histone-like protein HU-α. RaiZ forms an RNA duplex with the ribosome-binding site of *hupA* mRNA, facilitated by ProQ, to prevent 30S ribosome loading and protein synthesis of HU-α. Similarities and differences between ProQ- and Hfq-mediated regulation will be discussed.

**Keywords** HU-α; ProQ; RaiZ; small RNA; translation inhibition
**Subject Categories** Microbiology, Virology & Host Pathogen Interaction; Protein Biosynthesis & Quality Control; RNA Biology
**The EMBO Journal (2017) 36: 1029–1045**

## Introduction

Many if not all organisms use small base-pairing RNAs to modulate mRNA expression at the post-transcriptional level (Gorski *et al*, 2017; Kunne *et al*, 2014). These regulatory pathways often rely upon a conserved RNA-binding protein, primary examples of which are Argonaute family members in the microRNA pathway of eukaryotes (Huntzinger & Izaurralde, 2011; Meister, 2013) and the Sm-like protein Hfq in prokaryotes (Bossi & Figueroa-Bossi, 2016; De Lay *et al*, 2013; Updegrove *et al*, 2016; Vogel & Luisi, 2011; Wagner & Romby, 2015). Intense work on these pathways over the past decade has revealed the existence of large post-transcriptional networks that affect almost every cellular aspect and rival the complexity of primary gene expression control at the level of transcription.

The Hfq pathway has been particularly well mapped in the Gram-negative model bacteria *Escherichia coli* (Melamed *et al*, 2016; Schu *et al*, 2015; Tree *et al*, 2014), *Salmonella enterica* (Holmqvist *et al*, 2016), *Vibrio cholera* (Papenfort *et al*, 2015a) and *Pseudomonas* (Sonnleitner *et al*, 2008), in which the protein serves two general functions: protecting the Hfq-associated small noncoding RNAs (sRNAs) from cellular nucleases and helping them to recognize their target mRNAs. Most Hfq-associated sRNAs bind to their targets near the site of translational initiation (Melamed *et al*, 2016; Waters *et al*, 2016) and, therefore, this class of riboregulators primarily repress protein synthesis through steric interference with 30S ribosome binding (Balbontín *et al*, 2010; Bouvier *et al*, 2008; Morita *et al*, 2006; Udekwu & Wagner, 2007). However, additional mechanisms of repression have been reported which include deposition of Hfq within the mRNA 5′ untranslated region (5′ UTR) (Desnoyers & Massé, 2012) as well as target destabilization by recruitment of endoribonuclease RNase E to the mRNA coding sequence (CDS) (Bandyra *et al*, 2012; Pfeiffer *et al*, 2009). Conversely, Hfq-associated sRNAs also regulate some mRNAs positively by at least two different mechanisms, outcompeting translation-incompetent structures in the mRNA 5′-region (Papenfort *et al*, 2015b; Soper *et al*, 2010) or increasing mRNA stability by masking RNase E cleavage sites (Fröhlich *et al*, 2013; Papenfort *et al*, 2013). The same Hfq-associated sRNA may use multiple seed regions (Coornaert *et al*, 2013; Lee & Gottesman, 2016; Sharma *et al*, 2011) and no fewer than four different mechanisms to control its full suite of target mRNAs (Feng *et al*, 2015). Hfq is important for these sRNA–mRNA interactions which are usually imperfect and cannot be efficiently formed without assistance (Moll *et al*, 2003; Moller *et al*, 2002; Sobrero & Valverde, 2012; Updegrove *et al*, 2015; Zhang *et al*, 2002). Importantly, Hfq only interacts with single-stranded regions of its ligands and, once the sRNA–mRNA duplex has been formed, it typically dissociates and is available to bind other sRNAs (Fender *et al*, 2010; Hopkins *et al*, 2011; Ishikawa *et al*, 2012).

The wealth of molecular insight gained for the Hfq network overshadows the fact that these sRNAs constitute only a third of the ~300 sRNAs that have been annotated in, for example, *Salmonella* (Colgan *et al*, 2016; Westermann *et al*, 2016). Moreover, growing evidence suggests that regulation by Hfq represents only a part of

1  RNA Biology Group, Institute of Molecular Infection Biology, University of Würzburg, Würzburg, Germany
2  Helmholtz Institute for RNA-based Infection Research (HIRI), Würzburg, Germany
  *Corresponding author. Tel: +49 931 3182 575; Fax: +49 931 3182 578; E-mail: joerg.vogel@uni-wuerzburg.de
  †These authors contributed equally to this work

post-transcriptional regulatory processes in bacteria. Many microbes, such as *Helicobacter* or *Mycobacterium,* lack an Hfq homologue altogether (Chao & Vogel, 2010; Sharma *et al*, 2010; Wagner & Romby, 2015), and in *Staphylococcus*, Hfq is lowly expressed and dispensable for mRNA regulation (Bohn *et al*, 2007; Romilly *et al*, 2012). Even in canonical Hfq-containing *E. coli* and *S. enterica*, a number of functional Hfq-independent sRNA species have been described. They include most *cis*-acting antisense RNAs (asRNAs), which employ extensive perfect base pairing to repress mRNAs encoded on the opposite strand (Georg & Hess, 2011; Thomason & Storz, 2010). Plasmid-encoded asRNAs often use highly specific RNA chaperones (e.g. FinO, Rom) to assist these functions, whereas chromosomally encoded asRNAs are traditionally believed to operate in a protein-independent manner (Wagner & Romby, 2015). There are several additional specialized RNA–protein complexes; for example, CRISPR RNAs rely on dedicated molecular machinery provided by Cas proteins (van der Oost *et al*, 2014) and Y-like sRNAs associate with Ro proteins and PNPase to assist the degradation of structured RNAs (Chen *et al*, 2013). Naturally, sRNAs that do not employ base-pairing interactions to perform their functions but sequester certain regulatory proteins are also usually Hfq independent (Babitzke & Romeo, 2007; Göpel *et al*, 2013; Wassarman & Storz, 2000). However, additional proteins, other than Hfq, that define their own large classes of sRNAs have remained unknown.

Recently, we applied Grad-seq (RNA-seq-coupled partitioning of the transcriptome by density gradient centrifugation) to visualize the biochemical structure of *Salmonella* Typhimurium's RNA ensemble according to their involvement in ribonucleoproteins (RNPs) (Smirnov *et al*, 2016). While cosedimentation with Hfq explains the behaviour of ~20% of sRNAs, many additional sRNAs are apparently involved in different RNPs. Using sRNAs of this latter class as baits, we subsequently identified protein ProQ as a common binding partner.

ProQ is a conserved abundant RNA-binding protein of the ProQ/ FinO family that is widely spread in α-, β- and γ-proteobacteria (Attaiech *et al*, 2016; Chaulk *et al*, 2010, 2011; Glover *et al*, 2015; Smirnov *et al*, 2016) and whose solution structure has recently been solved in *E. coli* (Gonzales *et al*, 2017). We have demonstrated that ProQ associates with several hundred cellular transcripts, including dozens of sRNAs and that this protein has a profound impact on bacterial gene expression and physiology. On average, ProQ-associated sRNAs tend to be more folded than Hfq-dependent sRNAs, suggesting that ProQ preferentially binds transcripts with extensive secondary structure. While some of these sRNAs are part of known and putative type I toxin–antitoxin systems or were implicated in mRNA regulation by earlier studies, most are of unknown function (Smirnov *et al*, 2016).

Here, we report the characterization of a ProQ-dependent sRNA and the associated molecular function of the protein. We show that the RaiZ sRNA (formerly known as STnc2090; Chao *et al*, 2012) is induced upon entry in stationary phase and that it acts in *trans* to downregulate the translation of the *hupA* mRNA, which encodes the α-subunit of the bacterial histone-like protein HU. RaiZ forms a base-pairing interaction with the *hupA* ribosome-binding site (RBS) to repress translation. ProQ has a double role in this regulation: (i) it is necessary for the intracellular stabilization of RaiZ, and (ii) it together with the RaiZ-*hupA* duplex prevents 30S ribosome loading. These results lay the foundation for a mechanistic exploration of target regulation by the new large class of ProQ-associated sRNAs.

# Results

## Biogenesis of the RaiZ sRNA by 3′ mRNA processing

RaiZ was initially identified as candidate sRNA STnc2090 in a screen for Hfq-associated transcripts in *Salmonella* Typhimurium (Chao *et al*, 2012). It originates from the highly conserved *raiA* gene (encoding a cold shock-inducible ribosome-inactivating protein) of which it covers the last third of the CDS and the entire 3′ UTR. The RaiZ RNA sequence is conserved in several enterobacteria that are closely related to *Salmonella* (Fig 1A).

Northern blot probing of *S.* Typhimurium total RNA samples showed that RaiZ is primarily expressed in the stationary phase ($OD_{600} > 2$) or in a growth medium that induces the *Salmonella* pathogenicity island-1 (SPI-1) and less in the exponential phase or under *Salmonella* pathogenicity island-2 (SPI-2)-inducing conditions (Fig 1B), in accordance with available global RNA-seq profiling data (Kröger *et al*, 2013). In both *S. enterica* and *E. coli*, we detected two major RaiZ species, a 160-nt form (RaiZ) and a 122-nt processed sRNA (RaiZ-S), with a cumulative abundance of up to 50–60 copies per cell (Fig EV1). However, there are no transcription start sites within the *raiA* CDS (Kröger *et al*, 2012), suggesting that RaiZ is produced by endonucleolytic cleavage of the *raiA* mRNA. The cleavage site in the parental *raiA* mRNA that yields RaiZ is A/U-rich (Fig 1A), suggesting it would be a good substrate for the major mRNA processing enzyme RNase E (Mackie, 2013). Indeed, while in wild-type *Salmonella* RaiZ is efficiently produced at both 28°C and 44°C, the *raiA* mRNA accumulates in a thermosensitive *rne-3071* mutant (Apirion & Lassar, 1978) upon shifting to the non-permissive temperature, and RaiZ is no longer produced (Fig 1C), which is also confirmed by our recent genomewide analysis of RNase E cleavage sites (Chao *et al*, 2017). This supports a model whereby RaiZ arises from RNase E-mediated mRNA turnover, similar to the biogenesis of the 3′-end-derived sRNAs CpxQ and SroC (Chao & Vogel, 2016; Miyakoshi *et al*, 2015a).

## RaiZ is a ProQ-dependent sRNA

Although RaiZ was initially identified through its co-purification with Hfq (Chao *et al*, 2012), it has now emerged as a top ligand of ProQ, showing high enrichment in previous RIP-seq data obtained with a chromosomally FLAG-tagged ProQ protein (Fig 2A) (Smirnov *et al*, 2016). In addition, ProQ has been shown to bind both the longer and the shorter RaiZ forms in the low nanomolar range, indicating a strong interaction (Smirnov *et al*, 2016). As shown in Fig 2B, a ProQ-RaiZ complex is formed with high specificity and is even not affected by the presence of a 500-fold excess of tRNA. Using single-strand-specific Pb(II) treatment and the double-strand-specific RNase V1, we probed the native structure and identified the ProQ-protected sites of RaiZ (Fig 2C and Appendix Fig S1). In good agreement with *in silico* predictions (see Materials and Methods), both RaiZ and RaiZ-S contain several structured regions, including a large domain with an internal loop and a small hairpin next to the

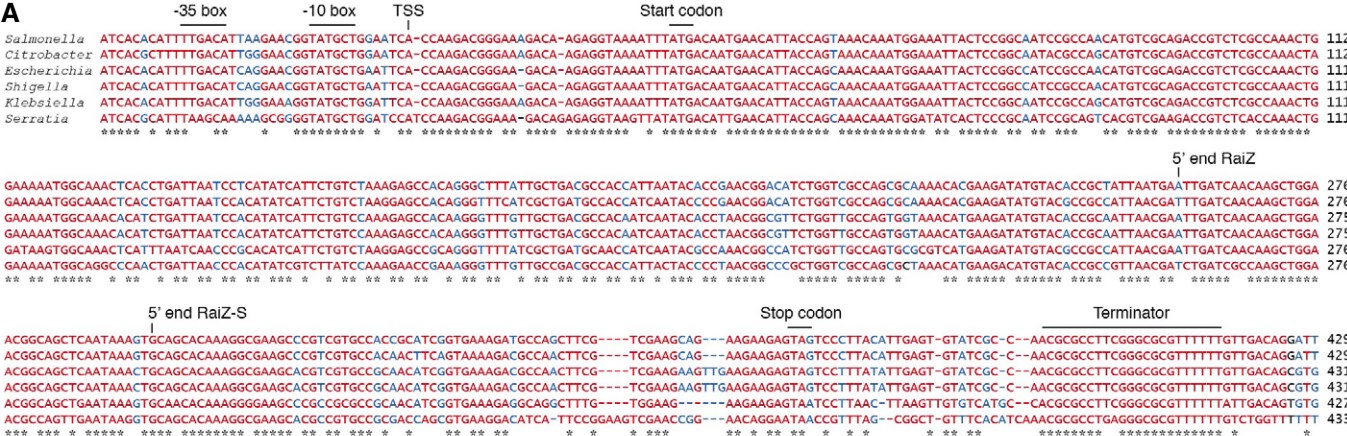

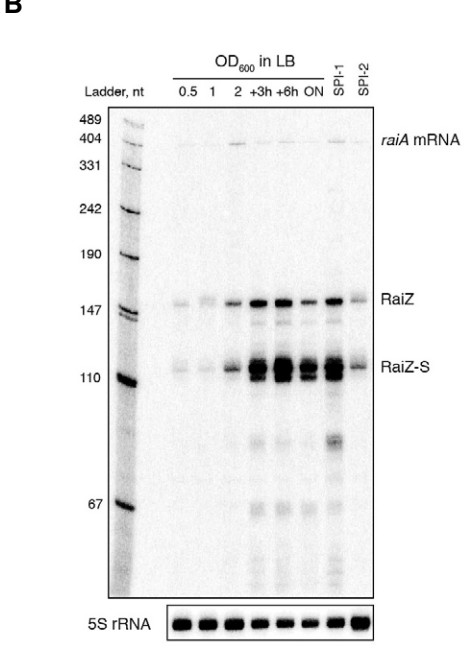

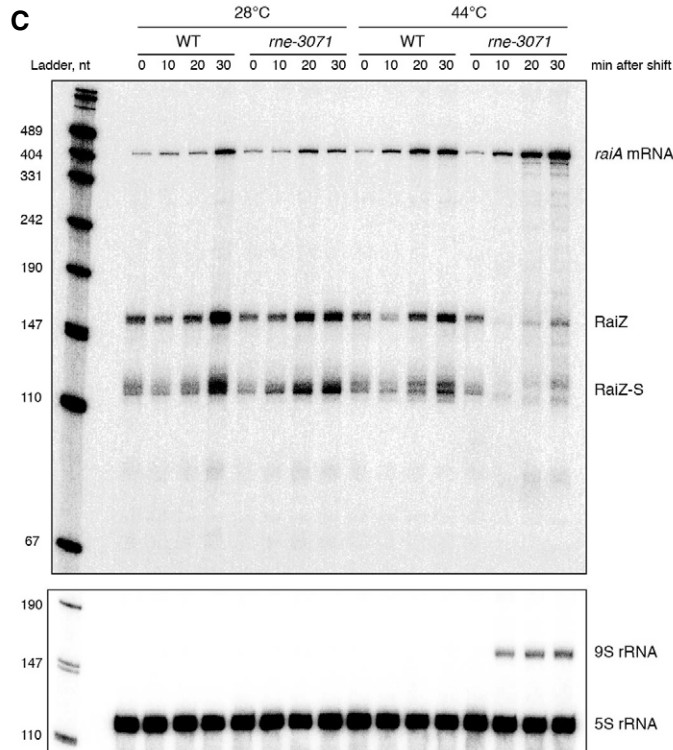

**Figure 1.  RaiZ is a processed enterobacterial sRNA.**

A  Multiple alignment of *raiA* loci from enterobacteria. Highly conserved positions are shown in red; invariant ones are marked with asterisks.

B  RaiZ expression in WT bacteria grown in LB or in SPI-1- and SPI-2-inducing media was visualized by northern blotting.

C  RNase E inactivation compromises the *raiA* mRNA processing and RaiZ production. Unlike the WT allele, the thermosensitive *rne-3071* variant gives rise to an RNase
   E protein which is only active at temperatures below 37°C, as can be assessed by the characteristic accumulation of a 5S rRNA precursor, 9S RNA, upon a shift to a
   non-permissive temperature of 44°C (Apirion & Lassar, 1978).

Source data are available online for this figure.

intrinsic terminator, separated by a long unstructured central spacer (Fig 2D). ProQ protects primarily the two 3′-terminal stem-loops and the base of the large 5′-terminal structured domain. These binding preferences resemble those of the protein FinO which is a well-characterized plasmid-encoded homologue of ProQ that interacts with the base of a stem-loop and the adjacent single-stranded

regions of the FinP sRNA (Arthur *et al*, 2011). Moreover, a *Legionella* ProQ homologue, RocC, also appears to recognize the Rho-independent terminator of its major target, the RocR sRNA (Attaiech *et al*, 2016). This binding mode is also in agreement with our recent analysis of the ProQ *in vivo* interactome which shows that ProQ strongly prefers structured RNAs (Smirnov *et al*, 2016).

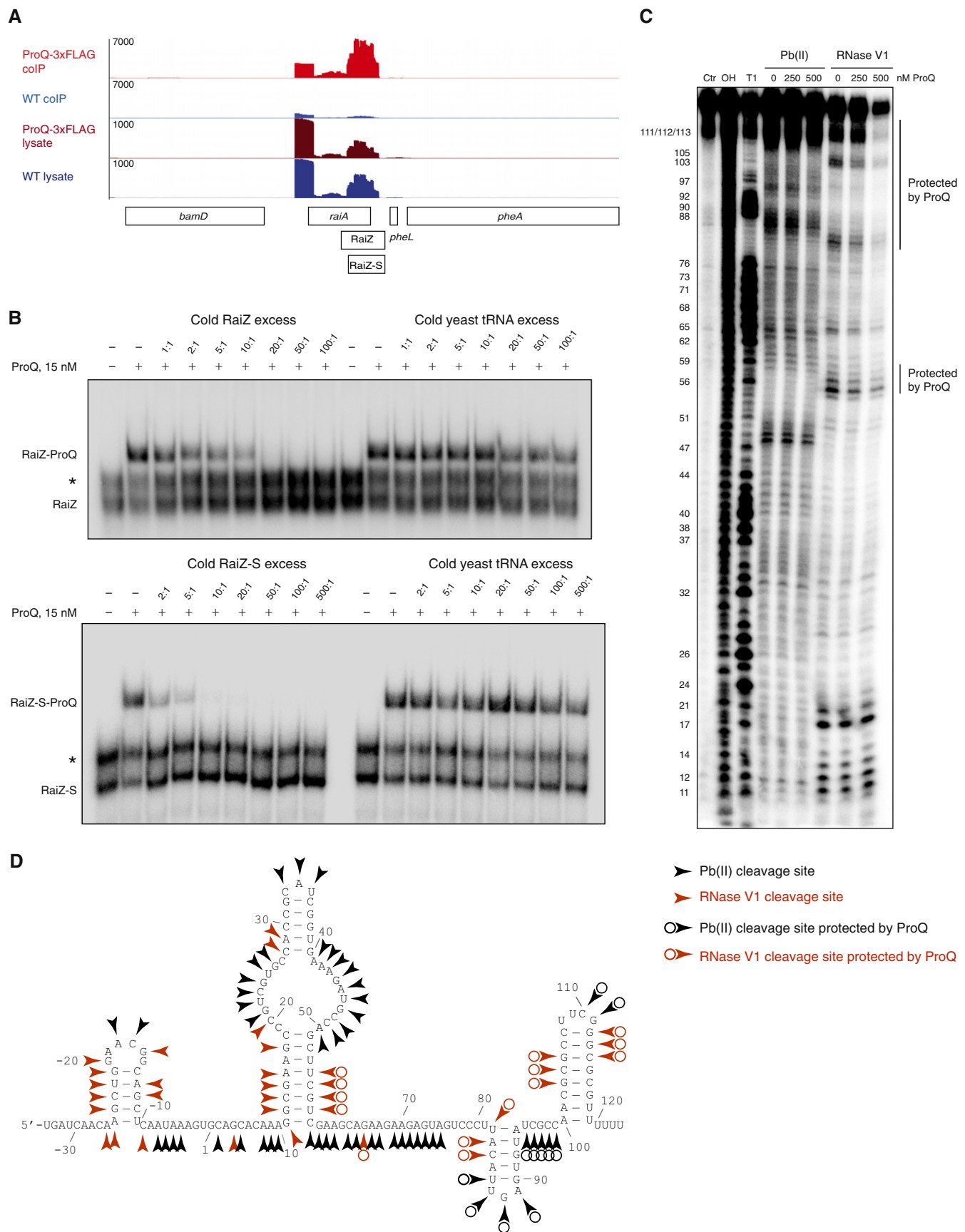

**Figure 2.**

◀

**Figure 2. RaiZ is a ProQ-binding sRNA.**

A   The read distribution around the *Salmonella raiAZ* locus for a ProQ-3xFLAG RIP-seq experiment performed in the transition phase (Smirnov *et al*, 2016). The upper two lanes show coIP fractions obtained by immunoprecipitation with anti-FLAG antibodies from a *proQ-3xFLAG* and a control WT strain without a tag; the lower two lanes show the corresponding total cell lysates. The linear scale (number of reads) is shown on the left. All genes are on the same (+) strand. Representative of four independent experiments.

B   RaiZ/RaiZ-S specifically interacts with ProQ. Competition experiments were carried out in the presence of either specific (cold RaiZ or RaiZ-S) or nonspecific (yeast tRNA) competitors. Asterisks mark RaiZ or RaiZ-S dimers observed under these conditions. Representative of two independent experiments.

C   *In vitro* footprinting assay of the RaiZ-S/ProQ complex. RaiZ-S is 5′-labelled. Ctr, uncleaved RNA; OH, alkaline ladder; T1, RNase T1 ladder. Nucleotide positions are shown on the left. Representative of two independent experiments. See also Appendix Fig S1 for the footprinting assay on the long form of RaiZ.

D   The secondary structure of RaiZ, based on the *RNAfold* prediction and the structure probing data shown in (C) and Appendix Fig S1. The first nucleotide of RaiZ-S is "1".

Source data are available online for this figure.

    In line with earlier observations showing that RaiZ efficiently interacts *in vivo* with Hfq (Chao *et al*, 2012; Smirnov *et al*, 2016), we confirmed the formation of a stable RaiZ-Hfq complex *in vitro* (Fig EV2). Therefore, RaiZ was found to engage in strong interactions with both ProQ and Hfq *in vitro* and *in vivo* (Figs 2 and EV2), which prompted us to evaluate the impact of each RNA chaperone on RaiZ stability. RaiZ was equally well produced in wild-type and Δ*hfq Salmonella*, but failed to accumulate in a Δ*proQ* strain (Fig 3A; Smirnov *et al*, 2016). Analysis of the RaiZ half-life in bacteria treated with rifampicin to arrest transcription clearly indicated that of the two RNA chaperones, only ProQ was required for RaiZ stability, whereas *hfq* deletion did not significantly affect the half-life of the sRNA (Fig 3B). The RaiZ stability defect in Δ*proQ* could not be rescued by over expression of the sRNA even from a high-copy plasmid (Fig EV3), indicating that ProQ primarily affects the half-life of RaiZ and not the transcription of *raiA*. Therefore, although both RNA chaperones bind RaiZ with high affinity, only ProQ was required for its stability.

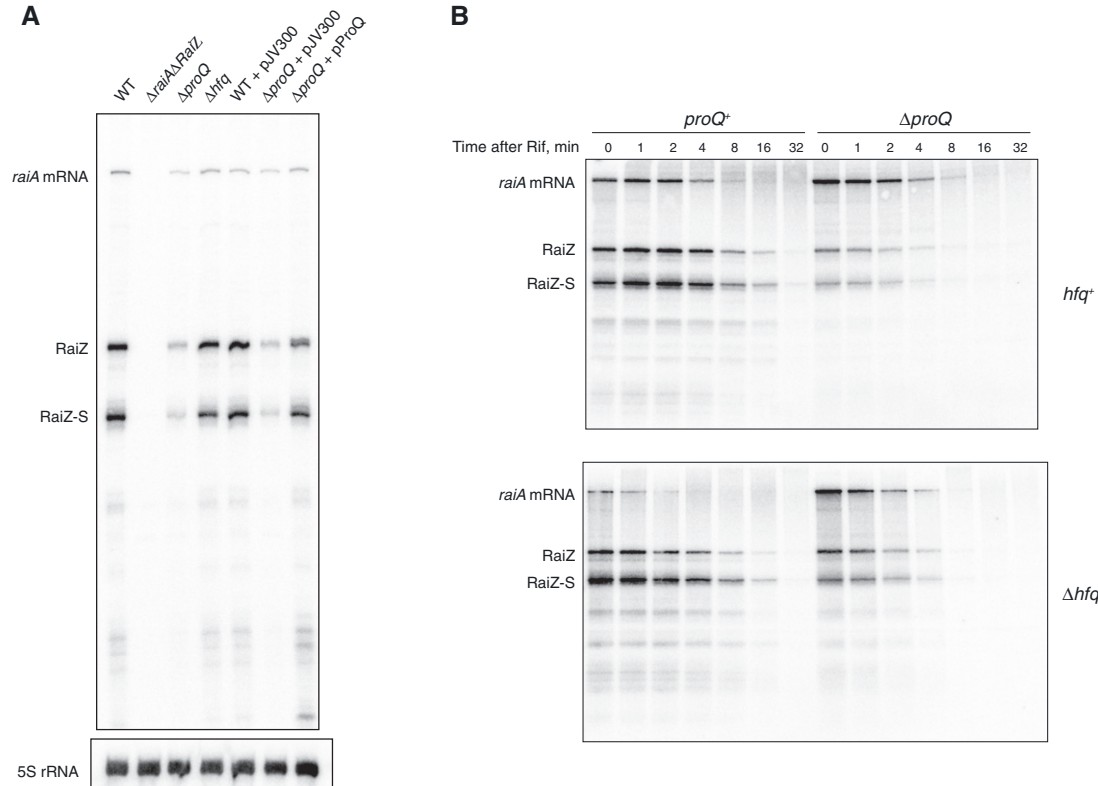

**Figure 3. RaiZ is a ProQ-dependent sRNA.**

A   Steady-state levels of RaiZ are compromised by *proQ* deletion but unaffected by *hfq* deletion. Total RNA from the corresponding strains was isolated at the transition phase and analysed by northern blotting. Δ*raiA*Δ*RaiZ* lacks the complete *raiA*-RaiZ locus, pJV300 is an empty control plasmid, and pProQ is a *trans*-complementing plasmid.

B   RaiZ stability was assessed in all four possible genetic backgrounds with respect to *hfq* and *proQ* genes. Cells were grown to the transition phase, rifampicin was added to arrest transcription, and total RNA samples were collected after the specified time intervals and quantified by densitometry after northern blotting.

Source data are available online for this figure.

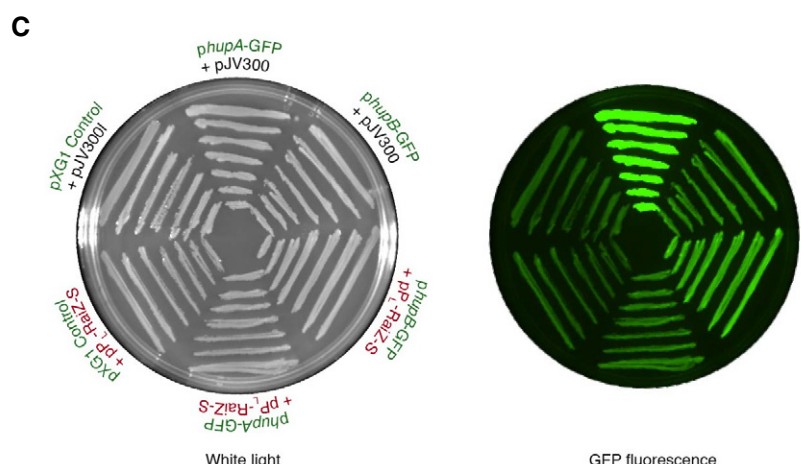

**Figure 4.  RaiZ negatively regulates *hupA* expression.**

A   Fold changes of *Salmonella* RNA levels 10 min after induction of RaiZ overexpression, as measured by RNA-seq of the total RNA. Differentially regulated genes (as compared to the control) are highlighted with colour. Data points correspond to mean genewise fold changes in two independent experiments, and the bars show the range.

B   Constitutive expression of RaiZ leads to the downregulation of HU-α production. Western (upper two panels) and northern blot (lower three panels) analyses were performed on total protein and RNA isolated from a *hupA-3xFLAGΔraiAΔRaiZ* strain constitutively expressing or not RaiZ.

C   RaiZ represses expression of a *hupA* fluorescent reporter construct. Constitutive expression of RaiZ specifically represses a *hupA*-GFP reporter containing the *hupA* 5′ UTR and the first 15 codons of the *hupA* CDS (constitutively expressed on a pXG10 plasmid), but does not affect a *hupB*-GFP reporter. Representative image from four independent experiments. See also Fig 6B for quantification of fluorescence in the same strains measured by FACS.

Source data are available online for this figure.

## RaiZ post-transcriptionally regulates *hupA*, encoding a histone-like protein

To obtain insight into the function of RaiZ, we performed a pulse-expression analysis (Massé *et al*, 2005; Papenfort *et al*, 2006) with the RaiZ sequence cloned into a multicopy plasmid under the control of an arabinose-inducible promoter. RaiZ expression was induced for 10 min in the exponential phase (when the chromosomal sRNA is barely expressed; Fig 1B), followed by RNA-seq to determine expression changes on the genomewide level. We observed a reproducible 6.9 ± 2.0-fold (mean ± SD) downregulation of a single mRNA encoding the α-subunit of the histone-like

protein HU, *hupA* (Fig 4A). Since the short time of induction makes secondary effects on gene expression unlikely (Sharma & Vogel, 2009), we considered *hupA* a direct target of RaiZ.

To validate this regulation, we overexpressed RaiZ under control of a constitutive promoter in a strain carrying a chromosomally FLAG-tagged allele of *hupA* (Fig 4B). In the control strain, the *hupA* mRNA accumulated almost exclusively in the exponential phase, whereas the corresponding protein levels remained constant throughout growth. In contrast, RaiZ overexpression affected the target at both the mRNA and protein levels, resulting in an 8.2 ± 2.9-fold (mean ± SD) decrease in HU-α production. These results were corroborated by the use of fluorescent reporter

constructs (Fig 4C). When a *gfp* CDS was cloned in frame with 15 N-terminal residues of HU-α preceded by the *hupA* mRNA 5′ UTR and under control of a constitutive promoter, overexpression of RaiZ resulted in significantly lower fluorescence, compared to a strain carrying the empty pJV300 plasmid. RaiZ overexpression did not affect a *gfp* reporter preceded by an unrelated 5′ UTR under the control of the same constitutive promoter (Fig 4C, pXG1) or an analogously constructed *hupB* reporter (see also Appendix Fig S2). These data indicate that of the two subunits of HU, encoded by *hupA* and *hupB*, only HU-α is subject to post-transcriptional regulation by RaiZ, and this regulation depended on its 5′ UTR and/or the start codon-proximal portion of the *hupA* CDS. In line with above data (Fig 3), this regulation was not affected by a Δ*hfq* mutation, suggesting that Hfq is not required for RaiZ-mediated *hupA* repression (Fig EV4).

### RaiZ is a base-pairing *trans*-acting sRNA

Since RaiZ efficiently repressed translation of a GFP reporter preceded by the *hupA* 5′ UTR and a few start codon-proximal codons (Fig 4C), we hypothesized that RaiZ may target the RBS of the *hupA* mRNA, as seen with many Hfq-dependent sRNAs (De Lay *et al*, 2013; Vogel & Luisi, 2011). Indeed, extensive though imperfect pairing, involving a total of 23 bases on either side and covering the upstream region of the start codon, was predicted between the two RNAs (Fig 5A and Appendix Fig S2). In agreement with this prediction, the two RNAs interacted efficiently *in vitro*, forming a duplex with apparent $K_d$ of ~80 nM (Fig 5B), which is similar to the affinity of other predicted ProQ-dependent sRNAs for which targets are known (Darfeuille *et al*, 2007; Ellis *et al*, 2015; Han *et al*, 2010; Silva *et al*, 2013; Smirnov *et al*, 2016).

Structure probing of the RaiZ-*hupA* mRNA duplex validated this targeting model (Figs 5C and EV5) and revealed several interesting features of the interaction. It showed a high degree of symmetry involving an upstream single-stranded region and a downstream stem-loop in both RNAs (Fig 5A). The stem-loop in each RNA formed base-pairing interactions with the opposite single-stranded stretch and the stem-loop of the partner, resulting in a long imperfect duplex. In RaiZ (both the long and the short forms), the sites concerned included the short hairpin upstream of the terminator and the adjacent portion of the long single-stranded spacer, whereas in the *hupA* mRNA they covered ~30 nucleotides of the 5′ UTR immediately adjacent to the start codon (Fig 5A). The perfectly base-paired central region of the duplex underwent a strong site-specific cleavage by RNase III *in vitro* (Figs 5C and EV5), indicative of an extensive and stable interaction. Interestingly, RaiZ-S conferred more efficient RNase III cleavage than the longer RaiZ (Fig EV5B), suggesting that the processed sRNA is particularly apt for the interaction and represents the active regulatory form of the sRNA.

To verify whether the downregulation of the *hupA* mRNA by RaiZ relies on the same interaction *in vivo*, we designed mutant versions of both partners by swapping two nucleotides engaged in the strongest stretch of the intermolecular duplex (Fig 5A). The mutant RNAs failed to form stable complexes with their wild-type partners and confer the characteristic strong RNase III cleavage in the correct position *in vitro* (Fig EV5A and B). As expected, ectopic expression of RaiZ under the control of a constitutive promoter in a *Salmonella* strain lacking the *raiA-RaiZ* locus demonstrated that

only the wild-type RaiZ and RaiZ-S were able to repress *hupA* expression. RaiZ and RaiZ-S containing the U81A, U82A mutations in the base-pairing region, failed to achieve a similar level of downregulation, despite accumulating to the same levels (Fig 6A). Interestingly, when using the full-size RaiZ construct, the RaiZ-S species accumulated, indicating that the long RaiZ form contains all structural elements necessary for correct RaiZ maturation.

When we used our GFP reporter system to assess the effect of these nucleotide substitutions on the *hupA* regulation *in vivo*, we again observed a significant decrease in fluorescence when both wild-type RaiZ and the *hupA* 5′ UTR-controlled *gfp* construct were co-expressed (Fig 6B). On the contrary, repression was completely relieved by mutations in either RaiZ (U81A, U82A) or the *hupA* 5′ UTR (A-11U, A-10U). Importantly, combination of both mutant partners, which restores base pairing, rescued wild-type levels of repression (Fig 6B). Altogether, these results prove that RaiZ downregulates *hupA* via a base-pairing interaction with its 5′ UTR near the RBS.

### ProQ assists RaiZ in preventing ribosome loading on the *hupA* mRNA

The RaiZ-*hupA* mRNA interaction occurs very efficiently and does not require assistance of either ProQ or Hfq *in vitro* (Fig 5B). Nevertheless, ProQ is critically required for RaiZ stability in the cell (Fig 3). To determine whether ProQ has a role in the RaiZ-dependent *hupA* repression beyond maintaining sRNA abundance, we overexpressed RaiZ-S in the *proQ*+ and Δ*proQ* backgrounds, which resulted in the saturation of the sRNA levels well beyond the apparent $K_d$ in both strains (Fig 7A; the estimated resulting RaiZ-S concentrations are > 4 μM, see Materials and Methods for further detail). Strikingly, while *hupA* expression was strongly repressed by RaiZ-S in the *proQ*+ strain, the sRNA failed to fully deplete HU-α in the absence of ProQ. Analogously, whereas during the transition phase ($OD_{600}$ = 2) *hupA* mRNA level dropped ~4-fold in the *proQ*+ strain compared to the same strain carrying the control plasmid, it remained constant at ~75% of control in the Δ*proQ* background (Fig 7A). Therefore, although ProQ has a major impact on RaiZ stability, these results suggest that ProQ may also be required for regulation downstream of RaiZ production and the RaiZ-*hupA* mRNA interaction.

Since RaiZ affects HU-α protein levels to a greater extent than the mRNA (Figs 6A and 7A), we hypothesized that it primarily interferes with translation, with mRNA destabilization being a secondary consequence of lower ribosome occupancy. Using the toeprint assay, we analysed the effect of RaiZ-S on 30S ribosome loading on the *hupA* translation initiation region (Fig 7B). In the presence of 30S subunits and formylmethionylated initiator tRNA, a characteristic strong toeprint was observed ~15 nt upstream of the start codon, indicating the correct assembly of the translation initiation complex. Addition of RaiZ-S resulted in a small but dose-dependent decrease of the toeprint signal, demonstrating that the sRNA is capable of interfering with 30S ribosome loading, albeit not efficiently. Strikingly, simultaneous addition of both RaiZ-S and ProQ resulted in the strong suppression of the toeprint, paralleled by the appearance of a new reverse transcriptase stalling site downstream, in front of the region involved in the base pairing with RaiZ (Fig 7B). This new signal did not depend on the presence of 30S subunits or tRNA and could not be produced by ProQ alone, suggesting that it corresponds

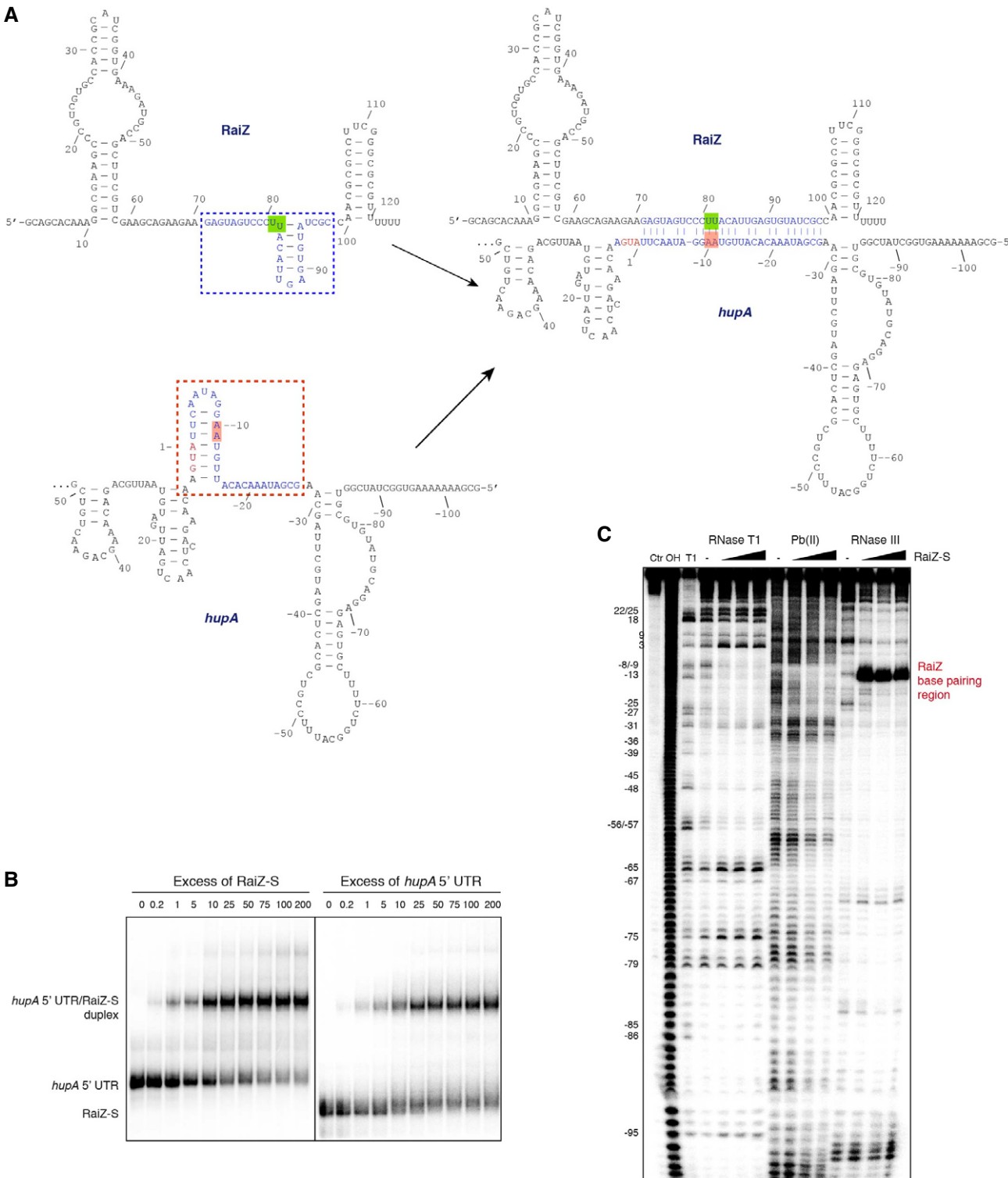

**Figure 5.  RaiZ base pairs with the RBS of the *hupA* mRNA.**

A   The RaiZ/*hupA* mRNA interaction, based on *RNAfold* and *RNAcofold* predictions and structure probing data in (C). The start codon is red, and A is numbered "1". The base-pairing regions are set in blue and framed. The sites where disruptive point mutations were introduced are highlighted with colour.

B   EMSA of the RaiZ/*hupA* mRNA interaction with either RNA labelled and a nonlabelled partner. Apparent $K_d$ of the complex is ~80 nM.

C   Structure probing assay of the RaiZ-S/*hupA* mRNA duplex. *hupA* 5′ UTR and the proximal part of the CDS are 5′-labelled. Nucleotide positions on the left correspond to the panel (A). Representative of two independent experiments.

Source data are available online for this figure.

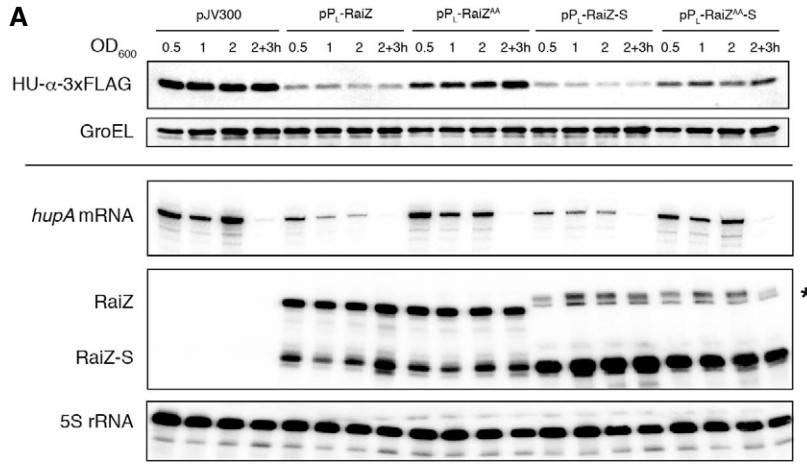

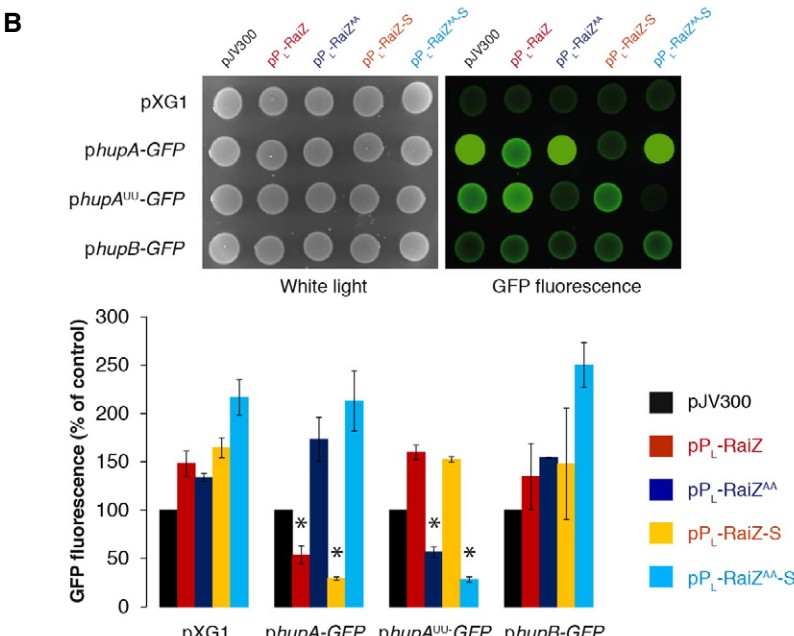

**Figure 6.  RaiZ-*hupA* mRNA base pairing is necessary for *hupA* repression.**

A  Constitutive expression of RaiZ leads to the downregulation of HU-α production only when the predicted base-pairing interaction is undisrupted. Western (upper two panels) and northern blot (lower three panels) analyses were performed on total protein and RNA isolated from a *hupA-3xFLAGΔraiAΔRaiZ* strain constitutively expressing or not RaiZ/RaiZ-S. RaiZ^AA^/RaiZ^AA^-S stand for the sRNAs carrying the double U81A, U82A substitution within the base-pairing region. Asterisk shows a read-through band coming from the expression vector. Representative of three independent experiments.

B  Constitutive expression of RaiZ/RaiZ-S specifically represses a *hupA*-GFP reporter, containing the *hupA* 5′ UTR and the first 15 codons of the *hupA* CDS (constitutively expressed on a pXG10 plasmid), but does not affect a *hupB*-GFP reporter. The RaiZ^AA^ mutation or the mirroring *hupA*^UU^ substitution (A-10U, A-11U) alleviates the repression when combined with WT partners, but they are fully compensated when combined with each other. Lower panel shows FACS quantification for three independent experiments (mean ± SD), *$P < 0.009$ (two-tailed Student's *t*-test, FDR-adjusted).

Source data are available online for this figure.

to a tripartite complex involving the *hupA* 5′ UTR, RaiZ-S and ProQ. Indeed, a stable ternary complex was observed in electrophoretic mobility shifts assay (EMSA) in the presence of ProQ (Fig 7C). Therefore, ProQ together with the RaiZ-*hupA* mRNA duplex may further prevent 30S ribosomes from loading onto and initiating translation of the *hupA* mRNA.

## Discussion

The vast majority of currently known ProQ-binding sRNAs are of unknown function (Smirnov *et al*, 2016). We have previously observed that asRNAs are enriched in the ProQ interactome, suggesting that this protein may be involved in gene expression

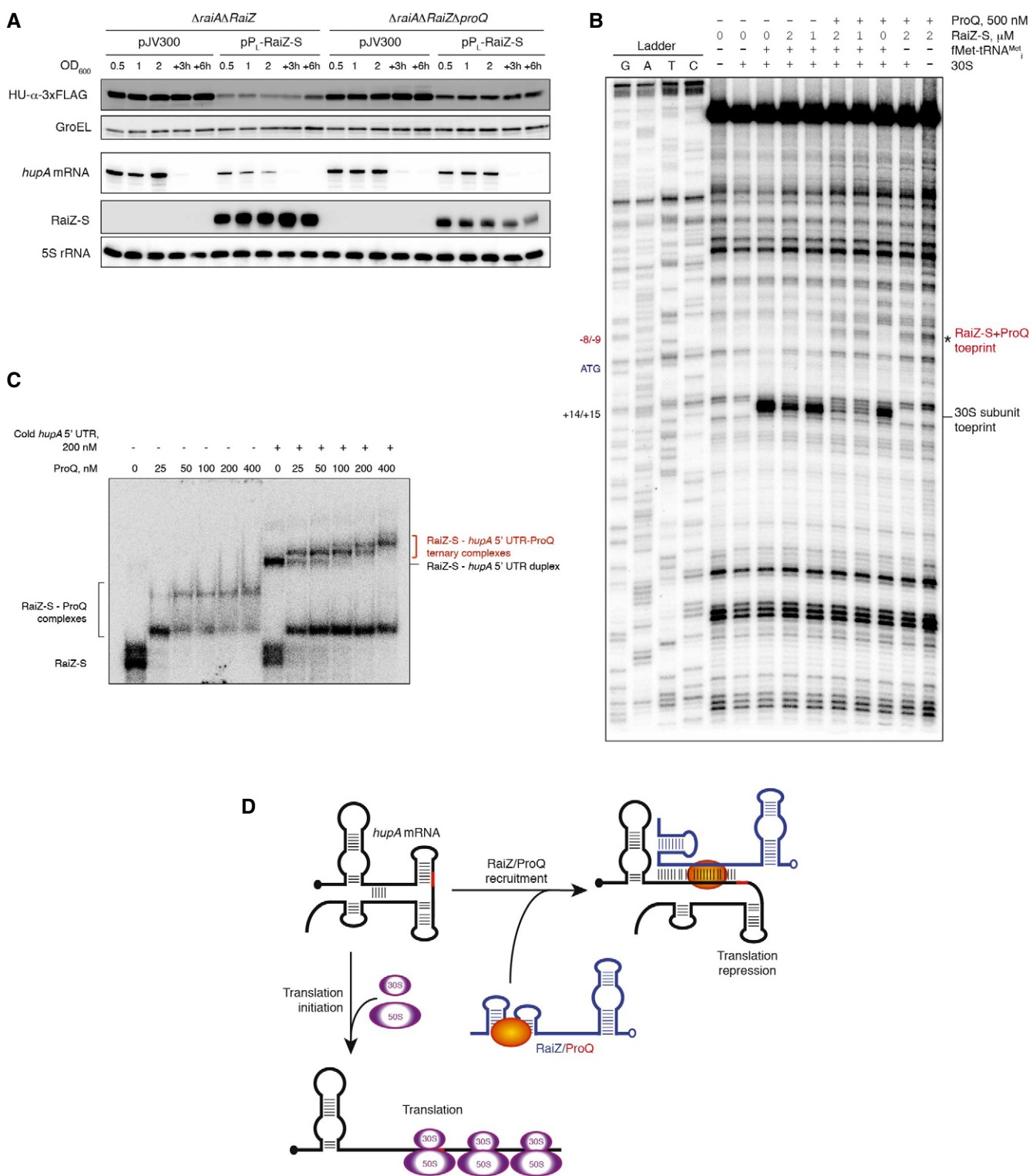

**Figure 7.**

regulation via perfect base pairing with *cis*-encoded mRNA targets. Some of these asRNAs and their regulatory mechanisms have been characterized, including members of the Sib, Rdl, and IstR families of type I antitoxins, the transposon-associated art200 and the intergenic

*cis*-acting SraG sRNAs (Darfeuille *et al*, 2007; Ellis *et al*, 2015; Fontaine *et al*, 2016; Han *et al*, 2010; Kawano, 2012; Mok *et al*, 2010). Some other ProQ-associated sRNAs are derived from transcriptional attenuators (SraF, *rimP* leader) or have been proposed to function as *trans*-encoded base-pairing sRNAs (SraL) (Argaman

**Figure 7.  Role of ProQ in the RaiZ/*hupA* mRNA regulatory interaction.**

A  Constitutive expression of RaiZ leads to the full repression of HU-α production only in the presence of a functional *proQ* allele. Western (upper two panels) and northern blot (lower three panels) analyses were performed on total protein and RNA isolated from a *hupA-3xFLAGΔraiAΔRaiZ* or a *hupA-3xFLAGΔraiAΔRaiZΔproQ* strain constitutively expressing or not RaiZ-S.

B  Toeprinting assay on the *hupA* mRNA performed in the presence or absence of RaiZ-S and/or ProQ. Numbers on the left correspond to the mRNA positions as in Fig 5A. Representative of two independent experiments.

C  RaiZ, *hupA* mRNA and ProQ form a stable ternary complex. EMSA experiments were carried out with ³²P-labelled RaiZ-S in the presence or absence of cold *hupA* 5′ UTR and increasing concentrations of ProQ. Note the appearance of supershifted bands in the presence of ProQ, corresponding to ternary *hupA*-RaiZ-ProQ complexes. The remaining RaiZ-S–ProQ complex observed in the presence of *hupA* 5′ UTR may represent a kinetically trapped, base-pairing-incompetent sRNA form. The *hupA* 5′ UTR alone shows only a modest, order of magnitude weaker affinity for ProQ, compared to RaiZ (see Appendix Fig S3).

D  The proposed model of ProQ-dependent RaiZ-mediated regulation of *hupA* mRNA translation. In the absence of the sRNA, the weak secondary structure of the RBS easily opens up and allows translation initiation. Upon RaiZ binding, the duplex is sealed by ProQ just upstream of the start codon, preventing the 30S ribosome from acceding the RBS and beginning translation.

Source data are available online for this figure.

*et al*, 2001; Naville & Gautheret, 2010; Nechooshtan *et al*, 2009; Plumbridge *et al*, 1985; Silva *et al*, 2013; Sittka *et al*, 2008). Recently, one of the ProQ-bound sRNAs derived from a tRNA trailer sequence (STnc2180 in *Salmonella*) has been shown to behave as a sponge to control the level and activity of other sRNAs in *E. coli* (Lalaouna *et al*, 2015). We have previously shown that ProQ stabilizes many of its sRNA ligands but its immediate role in their regulatory functions has not been addressed (Smirnov *et al*, 2016).

Here, we describe the first example of sRNA-mediated mRNA regulation involving ProQ in the model enterobacterium *Salmonella* Typhimurium. We show that RaiZ specifically base pairs with its unique target, the *hupA* mRNA, and, with the help of ProQ, inhibits translation initiation by interfering with 30S ribosome loading (Fig 7D). Our data suggest that ProQ plays a double role in this mechanism: (i) it critically enhances RaiZ stability, thus allowing a larger time window for its regulatory activity, and (ii) it binds the inhibitory complex involving both base-paired RNAs which helps to prevent 30S ribosome binding.

Interestingly, the regulatory mode adopted by RaiZ combines the properties of both Hfq-dependent *trans*-acting sRNAs and Hfq-independent *cis*-acting asRNAs. Similar to a significant fraction of Hfq-interacting sRNAs (Chao *et al*, 2012, 2017; Miyakoshi *et al*, 2015b), RaiZ is generated through a RNase E-dependent maturation process. Both RaiZ and RaiZ-S strongly bind to ProQ via their 3′ stem-loops but the latter form seems to be more functionally proficient, suggesting that the level of RaiZ regulatory activity may be controlled by processing, as it has been suggested for the Hfq-dependent ArcZ sRNA (Chao *et al*, 2017; Papenfort *et al*, 2009; Soper *et al*, 2010). Structurally, RaiZ resembles canonical Hfq-dependent sRNAs (Updegrove *et al*, 2015): An extensive single-stranded region, participating in the interaction with the *hupA* mRNA, is followed by a stem-loop and a typical intrinsic terminator. Yet, it lacks U-rich sequences near the seed, which have been shown to be important in Hfq-dependent regulatory modes (Ishikawa *et al*, 2012; Updegrove *et al*, 2015), and contains an additional structural domain that also contributes to ProQ binding. Therefore, although the general RaiZ design follows the same pattern as Hfq-dependent riboregulators, it is unlikely to function as such. Moreover, the base-pairing regions of RaiZ and the *hupA* mRNA show similarities to those of *cis*-encoded antisense RNAs (Fig 5A). The critical part of the RaiZ seed is folded into a short stem-loop which reciprocally contacts an analogous structural element on the mRNA side. Similar base-pairing patterns involving terminal loops have been documented in Sib/*isb* antitoxin–toxin loci (Han *et al*, 2010), the IS200

transposition-regulating art200/*tnp* module (Ellis *et al*, 2015), the F plasmid FinP/*traJ* conjugation-controlling system (Gubbins *et al*, 2003; Koraimann *et al*, 1996), and, most recently, in the RocR/*comEA* interaction which limits natural competence in *Legionella* (Attaiech *et al*, 2016).

The examples of FinP and RocR are of particular interest as these sRNAs function under control of the RNA chaperones FinO and RocC, respectively, which belong to the same protein family as ProQ, albeit they remain highly specialized with FinP and RocR being their only sRNA ligands (Attaiech *et al*, 2016; Glover *et al*, 2015). It is likely, therefore, that other sRNAs under control of ProQ/FinO proteins may employ similar base-pairing mechanisms relying on the inherent ability of these chaperones to deal with structured RNA substrates (Arthur *et al*, 2011; Smirnov *et al*, 2016). Interestingly, unlike most of the above-mentioned sRNAs, RaiZ and RocR act in *trans* engaging in imperfect base-pairing interactions, a trait traditionally associated with Hfq-dependent functions. This suggests that ProQ-dependent sRNAs may act on both *cis*- and *trans*-encoded targets while using similar base-pairing strategies.

The role of ProQ in early regulatory events involving RaiZ is reminiscent of that of Hfq, FinO or RocC. In all four cases, the RNA chaperone contributes to regulation by stabilizing its sRNA ligands. Deletions of *hfq*, *finO* and *rocC* are accompanied by significant depletion of Hfq-dependent sRNAs, FinP and RocR, respectively, leading to the deregulation of their target genes (Attaiech *et al*, 2016; Jerome *et al*, 1999; Vogel & Luisi, 2011). Interestingly, although RaiZ also co-immunoprecipitates with Hfq (Chao *et al*, 2012) and engages in a high affinity interaction with this RNA chaperone *in vitro*, deletion of *hfq* does not seem to affect RaiZ abundance, stability or sRNA-mediated *hupA* repression. By contrast, *proQ* deletion compromised RaiZ accumulation and resulted in an approximately fourfold drop in the sRNA stability (Fig 3; Smirnov *et al*, 2016), again suggesting that RaiZ functions primarily in a ProQ-dependent mode. The significance of RaiZ association with Hfq is currently unknown. Hfq may reside on Rho-independent terminators of transcripts without noticeable functional consequences (Sauer & Weichenrieder, 2011; Sittka *et al*, 2008) or, alternatively, take part in RaiZ processing. Interestingly, another ProQ-dependent sRNA, art200, was originally characterized as Hfq-associated (Sittka *et al*, 2008), although Hfq has no effect on either its stability or regulatory activity (Ellis *et al*, 2015). Therefore, the strong affinity for both RNA chaperones does not necessarily imply the involvement of a sRNA in both regulons. However, as the *E. coli* RaiZ orthologue was recently predicted to base pair with certain mRNAs as a part of Hfq-containing complexes

(Melamed *et al*, 2016), additional, Hfq-dependent, RaiZ-mediated regulatory mechanisms may exist.

Similar to many *cis*-encoded (Georg & Hess, 2011; Thomason & Storz, 2010) and some Hfq-dependent sRNAs (Desnoyers & Massé, 2012; Fei *et al*, 2015; Salvail *et al*, 2013), RaiZ does not seem to require a protein to facilitate base pairing with its target *in vitro* (Fig 5). Nonetheless, it fails to confer full repression on *hupA* in the absence of ProQ because the latter appears to be required for formation of a ternary complex which more efficiently prevents 30S ribosome loading (Fig 7). In this aspect, ProQ differs from Hfq which, being unable to stably associate with structured RNAs, typically cycles off the duplex once it has formed (Hopkins *et al*, 2011). This mechanistic feature appears to be unique among bacterial global RNA binders and likely stems from the inherent preference of ProQ for highly structured RNA substrates (Smirnov *et al*, 2016).

What is the rationale of the ProQ-dependent RaiZ regulation? We have previously demonstrated that ProQ interacts with and regulates several mRNAs for proteins involved in DNA-related processes (Smirnov *et al*, 2016). Intriguingly, a few ProQ-associated sRNAs have known or hypothetical roles in DNA pathways: IstR is induced in response to DNA damage (Vogel *et al*, 2004); art200 regulates IS200 transposition (Ellis *et al*, 2015); IsrB, IsrF, IsrJ, IsrK, and STnc1080 are prophage-encoded sRNAs with likely roles in lysogeny maintenance (Barquist *et al*, 2013; Hershko-Shalev *et al*, 2016; Padalon-Brauch *et al*, 2008); STnc40 is a hypothetical *mutT* attenuator (Naville & Gautheret, 2010; Pfeiffer *et al*, 2007); and STnc540 and STnc620 overlap the 3′ UTRs of the *himA* and *ssb* mRNAs encoding the α-subunit of the integration host factor and the single-stranded DNA-binding protein, respectively (Chao *et al*, 2012; Sittka *et al*, 2008). Of note, an *E. coli proQ* mutant has been predicted to be particularly sensitive to DNA-damaging agents in global screens (Nichols *et al*, 2011; Skunca *et al*, 2013). In *Legionella*, the ProQ homologue RocC plays a critical role in controlling competence (Attaiech *et al*, 2016; Sexton & Vogel, 2004). Therefore, regulation of HU-α, which is one of key DNA-binding proteins in bacteria with functions analogous to those of histones in Eukarya (Grove, 2011), by RaiZ contributes to a potential DNA maintenance-related ProQ regulon.

Interestingly, *Salmonella* RaiZ is produced in the stationary phase, upon oxygen limitation, oxidative stress, and under conditions inducing the *Salmonella* pathogenicity island 1 (SPI-1) expression (Fig 1C) (Kröger *et al*, 2013). It is known that in enterobacteria *hupA* expression decreases during growth, whereas *hupB* levels increase, resulting in a shift in the cellular HU composition from $\alpha_2$ homodimers to αβ heterodimers (Claret & Rouviere-Yaniv, 1997). The heterodimers are essential for maintaining *E. coli* viability in the stationary phase and cannot be functionally replaced by either homodimer (Claret & Rouviere-Yaniv, 1997; Oberto *et al*, 2009). Of particular relevance to our work, in *S*. Typhimurium HU deletions disparately impact the three large virulence, stress response and general physiology regulons (Mangan *et al*, 2011). This suggests that selective downregulation of *hupA* expression upon transition to the stationary phase is important for the remodelling of bacterial metabolism via impacting $\alpha_2$ and αβ HU forms, and the ProQ-dependent modulation of HU-α level by RaiZ may help mount an adequate transcriptional response to environmental insults while preserving the integrity of the bacterial genome.

Although the physiological function of RaiZ awaits further characterization, the current study provides a valuable model system for the investigation of the molecular functions of ProQ in a common model organism to dissect this new domain of post-transcriptional regulation in bacteria.

## Materials and Methods

### Bacterial strains and growth conditions

Detailed information on bacterial strains and their construction is listed in Appendix Table S1. *Salmonella* Typhimurium SL1344 was used throughout the study. Deletion mutants and chromosomally tagged derivatives were constructed as described (Datsenko & Wanner, 2000; Uzzau *et al*, 2001). All chromosomal modifications were transferred into a fresh *Salmonella* wild-type background by phage P22 transduction, and the $Km^R$ cassette was removed as described (Datsenko & Wanner, 2000). Unless specified differently, strains were grown at 37°C in Luria–Bertani (LB) medium supplemented with ampicillin (100 μg/ml), kanamycin (50 μg/ml) or chloramphenicol (20 μg/ml) when necessary. For growth under SPI-1-inducing conditions, single colonies were inoculated in 5 ml of LB with 0.3 M NaCl in tightly screwed 15-ml tubes and incubated for 12 h with vigorous shaking. Bacteria were grown at SPI-2-inducing conditions as described (Löber *et al*, 2006). Single colonies were inoculated in SPI-2 medium (80 mM MES, 4 mM tricine, 100 μM $FeCl_3$, 376 μM $K_2SO_4$, 50 mM NaCl, 1 mM $K_2HPO_4/KH_2PO_4$, 0.4% glucose, 15 mM $NH_4Cl$, 1 mM $MgSO_4$, 10 μM $CaCl_2$, 10 nM $Na_2MoO_4$, 10 nM $NaSeO_4$, 4 nM $H_3BO_4$, 300 nM $CoCl_2$, 100 nM $CuSO_4$, 800 nM $MnCl_2$, 1 nM $ZnSO_4$, pH 5.8) and grown overnight at 37°C with vigorous shaking. The overnight cultures were then diluted 50-fold with the same medium and grown in the same way for 12 h until $OD_{600}$ ~0.3.

### Oligonucleotides and plasmids

Oligonucleotides used for strain construction, cloning, *in vitro* RNA transcription template generation and northern blot hybridizations are listed in Appendix Table S2. Plasmids used in this study are summarized in Appendix Table S3.

### RNA extraction and northern blot analysis

Bacterial cultures were harvested and mixed with 0.2 volume RNA stop mix (5% acidic phenol in ethanol) and flash-frozen with liquid nitrogen. Total RNA was extracted with the TRIzol reagent (Invitrogen), and 10 μg RNA was analysed by northern blotting. RNAs were detected by hybridization with the 5′-labelled oligonucleotides, and autoradiographies were analysed by phosphorimager on a Typhoon FLA 7000 instrument with the help of the AIDA software (Raytest, Germany).

### RNase E *in vivo* cleavage assay

*Salmonella* RNase E thermosensitive strain JVS-07000 (*rne-3071 ts*, see Appendix Table S1) and an isogenic control strain JVS-06999 were grown overnight in LB medium at 37°C, then diluted 1:100 into fresh LB medium and further grown at 28°C to $OD_{600}$ of 2. Then they were either immediately transferred into a 44°C water bath or

left to grow at 28°C. At indicated time points, samples were collected and total RNA was isolated and analysed by northern blotting as described above.

## RNA half-life determination

Bacterial cultures were grown to an $OD_{600}$ of 2. They were then treated with rifampicin (final concentration of 500 μg/ml) to abrogate transcription. RNA samples were collected at indicated time points and quantified by northern blot analysis, as described above. The experiments were performed in biological triplicates.

## sRNA pulse expression and RNA-seq

Cultures of WT strains carrying either a pBAD-RaiZ plasmid or a control pBAD plasmid were grown to $OD_{600}$ of 0.5, when L-arabinose was added to a final concentration of 0.2%. RNA samples were collected at 0 and 10 min after induction, as described above. Total RNA was treated with DNase I (Fermentas), and cDNA library generation and sequencing were performed by Vertis Biotechnologie AG, Germany, as described (Sharma *et al*, 2010). The experiments were performed in biological duplicates.

## Protein level analysis by Western blotting and FACS

Overnight bacterial cultures were diluted 1:100 into fresh LB medium. At indicated time points, 0.1 OD cultures were harvested by centrifugation at 10,000 *g* for 2 min and cell pellets were resuspended in 100 μl 1× protein loading buffer (Fermentas) and incubated at 95°C for 5 min. After denaturation, 0.02 OD samples were resolved by SDS–PAGE and Western blotting was performed as described (Papenfort *et al*, 2013). The following antibodies and antisera were used: α-FLAG (Sigma, #F1804), α-GroEL (Sigma, #G6532). Signals were visualized with the Western Lightning reagent (PerkinElmer) and an ImageQuant LAS 4000 CCD camera (GE Healthcare). All experiments were performed in at least two biological replicates.

Strains CWS128–CWS147 (Appendix Table S1) were used for semi-quantitative GFP reporter plate assays. For this, 5 μl bacterial overnight cultures were plotted on LB agar plates by using 20 μl tips, dried at room temperature and incubated at 37°C for 12 h. Images were acquired by using the ImageQuant LAS 4000 CCD camera (GE Healthcare). For quantitative GFP FACS assay, 0.1 OD bacterial cells were harvested, washed twice with PBS and fixed with 4% paraformaldehyde. The GFP fluorescence intensity was quantified by flow cytometry with FACS Calibur (BD Bioscience). The experiments were performed in biological triplicates. Similar to what has been previously observed for other sRNAs such as DsrA, Spot42, RyhB, DapZ, OxyS, Yrl1 (Urban & Vogel, 2007; Chao *et al*, 2012; Hussein & Lim, 2011; Richter *et al*, 2010), GFP fluorescence in pXG-1-containing strains is higher when RaiZ is overexpressed, which is likely a nonspecific effect due to the saturation of cellular RNases resulting in the modest stabilization of mRNAs.

## *In vitro* synthesis and labelling of RNA

Transcripts used for *in vitro* assays (EMSA, RNA structure probing, 30S ribosome toeprinting) were synthesized with MEGAscript High

Yield Transcription Kit (AM1333, Ambion). DNA templates with T7 promoter sequences were generated by PCR with oligonucleotides listed in Appendix Table S2. RNA was isolated with phenol:chloroform:isopropanol (25:24:1), ethanol-precipitated at −80°C and gel-purified. For labelling, 20 pmol RNA was dephosphorylated with 10 units of calf intestine alkaline phosphatase (New England Biolabs) in a 20 μl reaction at 37°C for 1 h, followed by purification and precipitation, as above. The dephosphorylated RNA was 5′-labelled with 3 μl of $^{32}$P-γ-ATP (10 Ci/l, 3,000 Ci/mmol) and 1 unit of polynucleotide kinase (Fermentas) for 1 h at 37°C in a 20 μl reaction. Unincorporated nucleotides were removed with Microspin G-50 Columns (GE Healthcare), followed by purification of the labelled RNA on a denaturing polyacrylamide gel. Upon visualization of the labelled RNA with phosphorimager, the RNA band was excised from the gel and eluted with 0.1 M sodium acetate, 0.1% SDS, 10 mM EDTA at 4°C overnight, followed by phenol extraction and precipitation as before. Final concentrations were checked by NanoDrop 2000.

## Protein purification and electrophoretic mobility shifts assays

*Salmonella* Hfq and ProQ purification and EMSA were performed as described (Smirnov *et al*, 2016) with minor modifications. Briefly, native *Salmonella* Hfq and ProQ proteins were obtained by intein-based expression and purification (IMPACT). For EMSA, 5 nM of labelled RNA was used. Both labelled and unlabelled RNA were denatured at 95°C for 1 min and slowly cooled down to room temperature, then mixed together and/or with purified Hfq or ProQ protein in a final volume of 20 μl of 25 mM Tris–HCl, pH7.5, 150 mM KCl, 1 mM $MgCl_2$, and incubated at 25°C for 20 min. Reactions were resolved at 4°C on native 6–8% polyacrylamide (19:1) gels (in 0.5× TBE) at constant current of 50 mA for 3 h. Gels were dried and signals were analysed by phosphorimager as described above.

## RNA structure probing and footprinting assays

Structure probing and mapping of ProQ-binding regions were performed on *in vitro*-synthesized 5′-labelled RNA. Briefly, 0.2 pmol labelled RNA (in 5 μl) was denatured at 95°C for 1 min and chilled on ice for 5 min, followed by addition of 1 μl of 1 mg/ml yeast RNA (Ambion AM7118) and 1 μl of 10× structure buffer (0.1 M Tris–HCl, pH 7, 1 M KCl, 0.1 M $MgCl_2$). Unlabelled partner RNA or purified ProQ protein were then added at various ratios and incubated at 37°C for 10 min. Then, samples were either treated with 0.1 U RNase T1 (Ambion) for 3 min, or with 5 mM lead (II) acetate (Fluka) for 1.5 min, or with 1.3 U RNase III (New England Biolabs) for 6 min, or with 0.02 U RNase V1 (Ambion) for 5 min, or left untreated. Reactions were stopped by addition of 12 μl denaturing loading buffer (95% formamide, 18 mM EDTA, 0.025% SDS, 0.025% xylene cyanol, 0.025% bromophenol blue). RNase T1 sequencing ladders were prepared by using 0.4 pmol 5′-labelled RNA denatured at 95°C for 2 min in the presence of 1× structure buffer and chilled on ice; RNA was digested with 0.1 U RNase T1 for 5 min at 37°C. Alkaline (OH) sequencing ladders were prepared by incubating 0.4 pmol 5′-labelled RNA at 95°C for 5 min in the presence of alkaline hydrolysis buffer (Ambion). Reactions were stopped by addition of 12 μl denaturing loading buffer.

Samples were denatured prior to loading (95°C, 3 min) and separated on denaturing 6% sequencing gels containing 7 M urea in 1× TBE at constant power of 40 W. Gels were dried and signals were analysed by phosphorimager with the help of the AIDA software (Raytest, Germany), as described above.

### 30S ribosome toeprinting analysis

30S ribosome toeprinting assays were performed as described (Sharma *et al*, 2007) with few modifications. Briefly, 0.2 pmol of unlabelled *hupA* 5′ UTR-*gfp* fragment and 0.5 pmol of 5′-labelled JVO-01976 primer, complementary to the *gfp* fragment, were annealed together in annealing buffer (10 mM Tris acetate, pH 7.6, 1 mM DTT, 100 mM potassium acetate) for 1 min at 95°C and chilled for 5 min on ice. Then, magnesium acetate and NTPs were added to final concentrations of 10 and 0.5 mM, respectively. For inhibition analyses, 1 or 2 pmol of RaiZ-S was added. ProQ was added to a final concentration of 500 nM. All subsequent steps were carried out at 37°C. After 5 min of incubation, 2 pmol of 30S ribosomal subunits (gift from Dr. Knud Nierhaus, Max Planck Institute for Molecular Genetics, Berlin, Germany), pre-activated for 20 min, was added. After incubation for 5 min, uncharged tRNA$^{fMet}$ was added to a final concentration of 1 μM, and incubation continued for 15 min. Reverse transcription was carried out with 100 U Super-Script II (Invitrogen) for 20 min. cDNA synthesis was terminated with 100 μl of stop buffer (50 mM Tris–HCl, pH 7.5, 0.1% SDS, 10 mM EDTA), followed by phenol–chloroform extraction. Aqueous phases were added 5 μl of 3 M potassium hydroxide and incubated at 90°C for 5 min. After ethanol precipitation, cDNA was dissolved in 10 μl of loading buffer II (ThermoFisher Scientific). Sequencing ladders were generated with DNA Cycle Sequencing Kit (Jena Bioscience) according to the manufacturer's protocol on the same DNA template as used for T7 transcription and the same labelled primer as in the toeprinting reactions. cDNAs and sequence ladders were separated on a 6% polyacrylamide/7 M urea gel. Autoradiographs of dried gels were obtained as described above.

### Statistical tests

Sample size considerations and applicability of the *t*-test were verified following general guidelines described in Motulsky (2014). Conformity with a normal distribution was tested with the Anderson–Darling normality test, and *t*-test was performed with GraphPad QuickCalcs (www.graphpad.com/quickcalcs/).

**Expanded View** for this article is available online.

### Acknowledgements

The authors thank R. Reinhardt and K.U. Förstner for help with RNA-seq; T. Achmedov for technical assistance; F. Darfeuille, G. Wagner and S. Gorski for critical reading of the manuscript. This study was supported by Deutsche Forschungsgemeinschaft (DFG) grant Vo875/14-1 (to JV) and a Bavarian BioSysNet grant.

### Author contributions

AS and JV conceived the project, AS and CW designed experiments, CW, LLD and AS performed experiments, AS and JV wrote the paper, and all authors edited the manuscript.

### Conflict of interest

The authors declare that they have no conflict of interest.

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
