## [Review Process File · The EMBO Journal]

Manuscript EMBO-2016-96127

Molecular mechanism of mRNA repression in trans by a ProQ-dependent small RNA

Alexandre Smirnov, Chuan Wang, Lisa Drewry and Jörg Vogel

Corresponding author: Jörg Vogel, University of Würzburg

Review timeline:	Submission date:	20 November 2016
	Editorial Decision:	15 December 2016
	Revision received:	05 February 2017
	Accepted:	10 February 2017

Editor: Anne Nielsen

Transaction Report:

1st Editorial Decision

15 December 2016

Thank you for submitting your manuscript for consideration by the EMBO Journal. It has now been seen by three referees whose comments are shown below.

As you will see from the reports, all referees express high interest in the findings reported in your manuscript and praise the quality of the experimental work. However, they also raise a number of minor concerns that you will have to address before publication of the manuscript. Many of these points can be clarified via additional discussion and text revision but you will also need to include some additional experimental work.

Given the referees' positive recommendations, I would like to invite you to submit a revised version of the manuscript, addressing the comments of all three reviewers. I should add that it is EMBO Journal policy to allow only a single round of revision, and acceptance of your manuscript will therefore depend on the completeness of your responses in this revised version.

REFeree REPORTS

Referee #1:

In this very complete and carefully-prepared manuscript, Smirnov et al. report that the RaiZ small RNA (sRNA), which is derived from the 3' UTR of the *raiA* mRNA, relies on the ProQ RNA chaperone to base pair and repress the translation of the mRNA encoding the histone-like protein HU-a, thus providing the first example of a ProQ role in sRNA function in Enteric bacteria.

I primarily have editorial comments:

1. The authors should discuss the possible physiological consequences of RaiZ repression of HU- α but not HU- β synthesis as cells enter stationary phase.

2. The authors need to be a little more careful about wording with respect to three issues:

--The authors should be more cautious about their conclusion that RaiZ functions exclusively in a ProQ-dependent mode. Hfq does seem to have some effect on RaiZ stability in Figure 3B (though the levels are a little difficult to compare since the wild type and Δ hfq samples are not in the same panel). In addition, recent results from Melamed et al. (Mol Cell 63:884), suggest that Hfq facilitates base pairing between RaiZ and some mRNAs in *E. coli*.

--In several figures such as Figures 4B and 6B, it appears that the RaiZ-S is more active than RaiZ (as the authors also note). Thus, it might be better to use the term RaiZ-S when referring to the active sRNA.

--It is quite difficult to completely delineate the effects of ProQ on RaiZ stabilization versus prolonging base pairing in vivo. For example, in Figure 7A, the RaiZ-S levels are still higher in the wild type background. The authors are encouraged to be more guarded in how they describe these findings.

3. More explanation needs to be provided for a few figures:

--Figure 1B: The conditions and observations for the last SPI-2 lane should be mentioned somewhere.

--Figure 4C: It would be nice if the authors could provide quantitation for the fluorescence.

--Figure 6: The authors should provide a key for the graph in panel B (ideally in the figure), given that colors of the labels in the top panel and the colors of the bars do not match precisely (especially for the yellow bar). They should also provide an explanation for why the levels of fluorescence vary ~2-fold for the pXG1-containing strains.

4. Some sentences are confusing and should be edited for clarification:

--Page 5, end of first paragraph: There are two "recently" in a row.

--Page 6, line 12: "it together with the RaiZ-hupA duplex it prevents..."

--Page 10, bottom line: "...which is similar to the affinities of other sRNAs later found to be ProQ-dependent"

--Page 11, line 12: "The perfectly base-paired central portion..."

--Page 11, last two lines: "As expected, ectopic expression of RaiZ/RaiZ-S constructs..."

--Page 13, first four lines: "...RaiZ failed to fully deplete...it remained constantly 75%..."

--Page 15, line 14: "Given that 3'UTR-derived sRNAs..." This sentence seems disconnected from the rest of the paragraph.

--Page 33, line 6: "backgrounds relative the hfq".

Referee #2:

This manuscript presented for the first time a detailed mechanistic study on ProQ, a specific RNA-binding protein that has been recently discovered as a main player of sRNA-dependent regulation in *Salmonella Typhimurium*. ProQ has been found associated with several structured sRNAs although its recognition site and mechanism of action are still poorly defined. In this report, the authors presented series of well done experiments both in vivo and in vitro and showed the following data: (1) A small non coding RNA RaiZ arising from the processing of a large 3'UTR of *raiA* is able to repress the translation of *hupA* mRNA encoding the histone-like protein HU- α . (2) In vitro, ProQ protects a large part of the 3' end of RaiZ against Pb(II)-induced cleavages and RNase V1 hydrolysis. (3) In vivo, ProQ stabilizes RaiZ maintaining a sufficient level of the sRNA for efficient regulation. (4) ProQ is required to efficiently occlude the ribosome loading on *hupA* mRNA in the presence of RaiZ. (5) Hfq binds to RaiZ but this interaction has no biological consequence on RaiZ stability nor RaiZ-dependent repression of *hupA* mRNA translation. This manuscript highlights the complexity of intricate interactions between sRNAs and RNA-binding proteins for specific and efficient regulation. Even more interestingly is the hypothesis presented in the discussion suggesting that ProQ would contribute to preserve the integrity of genomes in response to various stresses,

giving rise to specialized sRNPs.

Minor suggestions:

1- Figure 7C seems to indicate that two ProQ molecules bind to RaiZ. However, the data as shown in Figure 2B looks different. It is also intriguing that in the presence of the mRNA, a large proportion of RaiZ is still bound to ProQ and this complex seems to be not competent for mRNA binding. The authors should briefly comment on that.

2- Why the footprinting assays (Figure 2C) have been performed with high concentrations of ProQ (1 and 2 μ M)? It is strange that the amount of full-length RaiZ decreases with increasing concentrations of ProQ. I would like to be sure that the strongest protections (at the top of the gel) resulted from ProQ binding and not from an artifact due to the aggregation of the RNA into the pocket. The protections around positions 26 to 30 are very weak and not significant. The protections observed at the first hairpin motif of RaiZ are not shown in Figure 2C. It should also be indicated in the legend of the figure that the numbering of the nucleotides corresponding to RaiZ-S. For clarity, it would be better to represent on the secondary structure model the cleavages and the effect of the protein binding.

3- Does ProQ bind to hupA mRNA?

4- The fact that the binding of RaiZ to Hfq has no biological relevance is quite unexpected. The data presented here cannot rule out that other targets of RaiZ might require Hfq instead of ProQ.

Referee #3:

In "Molecular mechanism of mRNA repression in trans by a ProQ-dependent small RNA", Smirnov and colleagues define the molecular mechanism by which ProQ, a recently discovered sRNA binding protein, controls the function of a sRNA, RaiZ. Previously, the authors discovered RaiZ as a sRNA that interacts with ProQ. In this manuscript, they first define the expression pattern and origin of RaiZ, which is produced by processing of the *raiA* mRNA. They use biochemical methods to reveal the nucleotide bases important for RaiZ to interact with ProQ. They also demonstrate that ProQ is crucial for RaiZ stability. To further define the role of RaiZ in the cell, they perform RNA-seq to identify its targets. This screen produced *hupA* as a target of RaiZ. Companion biochemical analyses are used to define the nucleotide bases involved in the sRNA-mRNA interaction. Finally, the authors examine the functional role of ProQ in this regulation, showing that ProQ assists RaiZ in blocking the ribosome from loading onto *hupA*. Notably, while RaiZ also binds the well-known chaperone Hfq and the mechanism of action of RaiZ is similar to that of traditional Hfq-binding sRNAs, the authors convincingly show that RaiZ is, in fact, acting in conjunction with ProQ.

This manuscript is of very high quality. The work is logical, the writing is clear, and the data are excellent. All of the claims made are extensively backed up by convincing data. The arguments the authors make are persuasive.

I have some minor suggestions for improvement:

1. Figure EV1 - The estimates of copy number of RaiZ/cell seem surprisingly high based on the band strength of the samples shown in the figure. Providing quantitation of these data might strengthen the case. Can the authors check the quantification or tell us how they arrived at their conclusions?

2. Figure 2C - The authors need to specify what the difference is between the solid and dashed lines to the right of the gel. The solid and dashed lines both have the annotation "protected by ProQ".

3. Figure 4C - The evidence that RaiZ-S does not affect the *phupB*-GFP fusion is difficult to interpret as the signal from the reporter is less strong than the no-GFP control. It is possible this is an artifact of the image, but if that is the case, a quantitative measurement would be better. I wonder if the labeling of the figure panels was accidentally exchanged? In both Fig EV4 and Fig 6B, the *phupB*-GFP expression is stronger than the control. Is there a labeling mistake? If not, the authors need to explain these data.

We wish to thank all the referees for their enthusiasm regarding our manuscript and useful suggestions. Below we present a point-by-point response to their comments.

Referee #1:

In this very complete and carefully-prepared manuscript, Smirnov et al. report that the RaiZ small RNA (sRNA), which is derived from the 3' UTR of the *raiA* mRNA, relies on the ProQ RNA chaperone to base pair and repress the translation of the mRNA encoding the histone-like protein HU- α , thus providing the first example of a ProQ role in sRNA function in Enteric bacteria.

I primarily have editorial comments:

1. The authors should discuss the possible physiological consequences of RaiZ repression of HU- α but not HU- β synthesis as cells enter stationary phase.

Reply: The physiological consequences of RaiZ regulation of HU- α will likely be mediated through HU dimer formation. The two HU subunits, HU- α and HU- β , interact to form homo- and heterodimers, which have different cellular roles (Pinson et al, *J Mol Biol* 1999). During growth *hupA* mRNA levels progressively decrease and *hupB* levels increase (Claret & Rouviere-Yaniv, *J Mol Biol* 1997), consequently, the HupA₂ homodimer predominates in the exponential phase and the HupAB heterodimer in the stationary phase. The homo- and heterodimers are not functionally equivalent, for example, the HupAB heterodimer is essential for *E. coli* viability in the stationary phase and cannot be replaced by either homodimer (Claret & Rouviere-Yaniv, *J Mol Biol* 1997). Transcriptomic studies have also revealed that HupAB and HupA₂ differentially regulate genes in a variety of stress conditions (Oberto et al, *PLoS One* 2009) and HU subunit deletions disparately impact three large regulons responsible for virulence, stress response and general physiology in *S. Typhimurium* (Mangan et al, *Microbiology* 2011). Overall, this suggests that the selective downregulation of *hupA* expression upon transition to the stationary phase is important for the remodelling of bacterial metabolism via differentially regulating HupA₂ and HupAB HU forms. Based on the suggestion of the Referee, we now develop this point in more detail in Discussion (p18-19).

2. The authors need to be a little more careful about wording with respect to three issues:

--The authors should be more cautious about their conclusion that RaiZ functions exclusively in a ProQ-dependent mode. Hfq does seem to have some effect on RaiZ stability in Figure 3B (though the levels are a little difficult to compare since the wild type and Δ hfq samples are not in the same panel). In addition, recent results from Melamed et al. (*Mol Cell* 63:884), suggest that Hfq facilitates base pairing between RaiZ and some mRNAs in *E. coli*.

Reply: We are grateful to the reviewer for highlighting these important issues. Hfq likely binds to the RaiZ terminator, which would explain the detection of RaiZ-mRNA chimaeras by RIL-seq (Melamed et al, *Mol Cell* 2016). Although it is currently unclear whether these RaiZ chimaeras with other *E. coli* mRNAs (observed by RIL-seq) represent bona fide sRNA-mediated regulation of additional targets, we agree that this possibility should be mentioned and that a more cautious explanation regarding the relationship of RaiZ and Hfq is warranted. We have moderated the corresponding conclusions accordingly (p10, 16-17) and further discussed the results by Melamed et al, 2016 in the text (p17).

--In several figures such as Figures 4B and 6B, it appears that the RaiZ-S is more active than RaiZ (as the authors also note). Thus, it might be better to use the term RaiZ-S when referring to the active sRNA.

Reply: Based on the sum of our experiments, we similarly suspect that RaiZ-S is the active form of the sRNA in *Salmonella*. However, in the absence of direct evidence for this (which would require the generation of a processing-incompetent form of RaiZ), we prefer to avoid a more strongly worded conclusion. In addition, a survey of RNA-seq datasets available in the lab suggests that the ratio of RaiZ and RaiZ-S varies amongst species such that some bacteria express primarily the long form. Therefore, we have decided to use the term RaiZ-S to describe the results of experiments involving the short form, while using the more general term RaiZ when differences between the longer and the shorter forms could not be definitively established (e.g. when long RaiZ expression leads to RaiZ-S generation by processing) or do not appear to be significant.

--It is quite difficult to completely delineate the effects of ProQ on RaiZ stabilization versus prolonging base pairing in vivo. For example, in Figure 7A, the RaiZ-S levels are still higher in the wild type background. The authors are encouraged to be more guarded in how they describe these findings.

Reply: We agree with the Referee that separating these two effects is difficult and, although the levels of overexpressed RaiZ should be saturating in this experimental setup, we cannot formally rule out that the differential stability of RaiZ-S in the *proQ*⁺ and Δ *proQ* backgrounds may affect *hupA* regulation. We have moderated our conclusions accordingly (p13).

3. More explanation needs to be provided for a few figures:

--Figure 1B: The conditions and observations for the last SPI-2 lane should be mentioned somewhere.

Reply: The SPI-2-inducing conditions are now described in more detail in the Materials and Methods section (p20) and the observed lower expression of RaiZ is mentioned in the text (p7).

--Figure 4C: It would be nice if the authors could provide quantitation for the fluorescence.

Reply: We provide quantification of fluorescence for the strains in the manuscript measured by FACS, which is shown in Fig 6B. The semi-quantitative plate assay shown in Fig 4C is meant to provide the reader with an intuitive visualisation of the initial validation of the predicted *hupA* regulation by RaiZ. To clarify this point, we now refer the reader in the legend for Fig 4C to Fig 6B for precise quantification. We also clarify in the Materials and Methods (p22) that the plate assay was used as a semi-quantitative approach whereas FACS measurement was applied to the same strains for exact quantification of fluorescence.

--Figure 6: The authors should provide a key for the graph in panel B (ideally in the figure), given that colors of the labels in the top panel and the colors of the bars do not match precisely (especially for the yellow bar). They should also provide an explanation for why the levels of fluorescence vary ~2-fold for the pXG1-containing strains.

Reply: We have now included the key and hope that it improves the clarity of presentation. The reason for the observed sRNA-dependent variability is currently unclear. However, a similar nonspecific effect has previously been observed upon overexpression of other sRNAs such as DsrA,

Spot42, RyhB, DapZ, OxyS, Yrf1 by us and others (Urban et al, 2007, Chao et al, 2012, Hussein & Lim, 2011, Richter et al, 2010). Importantly, this does not affect the conclusion because this nonspecific effect on the *gfp* control is taken into account when calculating target regulation. We now briefly comment on this in the Materials and Methods (p22).

4. Some sentences are confusing and should be edited for clarification:

--Page 5, end of first paragraph: There are two "recently" in a row.

--Page 6, line 12: "it together with the RaiZ-hupA duplex it prevents..."

--Page 10, bottom line: "...which is similar to the affinities of other sRNAs later found to be ProQ-dependent"

--Page 11, line 12: "The perfectly base-paired central portion..."

--Page 11, last two lines: "As expected, ectopic expression of RaiZ/RaiZ-S constructs..."

--Page 13, first four lines: "...RaiZ failed to fully deplete...it remained constantly 75%..."

--Page 15, line 14: "Given that 3'UTR-derived sRNAs..." This sentence seems disconnected from the rest of the paragraph.

--Page 33, line 6: "backgrounds relative the hfq".

Reply: We are grateful to the Referee for highlighting these issues. We have now edited these sentences and hope that the corrections make the text clearer.

Referee #2:

This manuscript presented for the first time a detailed mechanistic study on ProQ, a specific RNA-binding protein that has been recently discovered as a main player of sRNA-dependent regulation in *Salmonella Typhimurium*. ProQ has been found associated with several structured sRNAs although its recognition site and mechanism of action are still poorly defined. In this report, the authors presented series of well done experiments both in vivo and in vitro and showed the following data: (1) A small non coding RNA RaiZ arising from the processing of a large 3'UTR of *raiA* is able to repress the translation of *hupA* mRNA encoding the histone-like protein HU- α . (2) In vitro, ProQ protects a large part of the 3' end of RaiZ against Pb(II)-induced cleavages and RNase V1 hydrolysis. (3) In vivo, ProQ stabilizes RaiZ maintaining a sufficient level of the sRNA for efficient regulation. (4) ProQ is required to efficiently occlude the ribosome loading on *hupA* mRNA in the presence of RaiZ. (5) Hfq binds to RaiZ but this interaction has no biological consequence on RaiZ stability nor RaiZ-dependent repression of *hupA* mRNA translation. This manuscript highlights the complexity of intricate interactions between sRNAs and RNA-binding proteins for specific and efficient regulation. Even more interestingly is the hypothesis presented in the discussion suggesting that ProQ would contribute to preserve the integrity of genomes in response to various stresses, giving rise to specialized sRNPs.

Minor suggestions:

1- Figure 7C seems to indicate that two ProQ molecules bind to RaiZ. However, the data as shown in Figure 2B looks different.

Reply: We thank the Referee for having spotted this apparent discrepancy. This work was carried out over a period of two years, since performing those initial experiments (previous panels 2B, 2C) we have optimised the purification protocol to obtain a more active ProQ protein. Subsequent experiments including those shown in Fig. 7C were performed with a more active preparation of ProQ. We have also optimised the binding conditions compared to those described in our previous

paper, Smirnov et al, *Proc Natl Acad Sci USA* 2016, and which were also used in Fig 2B. We have now repeated the experiments shown on Fig 2B and 2C with the ProQ preparation used in Fig 7. We observe the same behaviour as previously described: yeast tRNA fails to outcompete RaiZ whereas specific competition with unlabelled RaiZ is very efficient and dose-dependent. These new data now replace the former panels 2B and 2C.

It is also intriguing that in the presence of the mRNA, a large proportion of RaiZ is still bound to ProQ and this complex seems to be not competent for mRNA binding. The authors should briefly comment on that.

Reply: As the Referee correctly points out, the RaiZ-ProQ complex can be observed even when the *hupA* mRNA is present in solution (Fig. 7C). We propose two possible, not mutually exclusive, explanations why the remaining ProQ-bound RaiZ does not convert into RaiZ-*hupA* duplexes. It may be kinetically trapped in a base pairing-incompetent conformation. Alternatively, ProQ may not significantly impact the on-rate of the RaiZ-*hupA* interaction which is largely determined by the relative concentrations of the two RNAs, as can be seen by comparison of lanes with and without ProQ. Following the Referee's request, we briefly comment on this in the figure legend.

2- Why the footprinting assays (Figure 2C) have been performed with high concentrations of ProQ (1 and 2 μ M)? It is strange that the amount of full-length RaiZ decreases with increasing concentrations of ProQ. I would like to be sure that the strongest protections (at the top of the gel) resulted from ProQ binding and not from an artifact due to the aggregation of the RNA into the pocket.

The footprinting assays required a relatively high concentration of labelled RaiZ to achieve sufficient intensity of bands while keeping the cleavage statistical, and they were performed in the presence of a particularly high excess of yeast tRNA to limit activity of the footprinting reagents. Consequently, this forced us to use an excess of ProQ to maximise the proportion of complexed RNA throughout the entire experiment. Moreover, as we mentioned above (see response to point 1), at that time we worked with a less active ProQ preparation, which further explains the need for higher protein concentrations. As described above, we have repeated these experiments with a fully active ProQ preparation at lower concentrations. This has considerably improved the yield of the full-size RNA while recapitulating the same protection pattern by ProQ. Therefore, we believe that the two 3'-proximal stem-loops of RaiZ likely represent a true ProQ-binding site. These new data now replace the former Fig 2C.

We should also mention that these ProQ binding sites in RaiZ are fully supported by global CLIP-seq data (*in vivo* RNA-protein UV crosslinking) for ProQ in *Salmonella* and *E. coli* that we are currently preparing for publication (Erik Holmquist and Jörg Vogel). Specifically, we identified the main region protected by ProQ in RaiZ-S in positions 72-106, according to the numbering shown in Fig 2D.

The protections around positions 26 to 30 are very weak and not significant.

Reply: We agree with this and do not highlight them anymore in the new version of the figure.

The protections observed at the first hairpin motif of RaiZ are not shown in Figure 2C.

Reply: The assignment of cleavage positions in the first hairpin shown in Fig 2D is based on similar footprinting experiments and RNAfold predictions performed with the longer form of RaiZ (now added as a new Appendix Fig S1). Since no significant protection of this hairpin by ProQ was seen

at lower protein concentrations, we think that it may represent a low-affinity site that is not biologically significant *in vivo*.

It should also be indicated in the legend of the figure that the numbering of the nucleotides corresponding to RaiZ-S. For clarity, it would be better to represent on the secondary structure model the cleavages and the effect of the protein binding.

Reply: We have implemented these suggestions and hope that they have improved the clarity of presentation.

3- Does ProQ bind to hupA mRNA?

Reply: We observed a relatively weak binding of the *hupA* mRNA in our EMSAs (apparent $K_d \sim 50$ nM, which is an order of magnitude higher than K_d for the RaiZ-ProQ complex, Smirnov et al, *Proc Natl Acad Sci USA* 2016). It is, therefore, unlikely that *in vivo*, in the presence of RNAs that bind to ProQ with higher affinity, the *hupA* 5' UTR would significantly bind to ProQ. We have included these data in Appendix Fig S3 and briefly discussed them in the legend for Fig 7C.

4- The fact that the binding of RaiZ to Hfq has no biological relevance is quite unexpected. The data presented here cannot rule out that other targets of RaiZ might require Hfq instead of ProQ.

Reply: As outlined in our response to Referee #1, we fully agree with this comment and have moderated our conclusions accordingly (p10, 16-17).

Referee #3:

In "Molecular mechanism of mRNA repression in trans by a ProQ-dependent small RNA", Smirnov and colleagues define the molecular mechanism by which ProQ, a recently discovered sRNA binding protein, controls the function of a sRNA, RaiZ. Previously, the authors discovered RaiZ as a sRNA that interacts with ProQ. In this manuscript, they first define the expression pattern and origin of RaiZ, which is produced by processing of the *raiA* mRNA. They use biochemical methods to reveal the nucleotide bases important for RaiZ to interact with ProQ. They also demonstrate that ProQ is crucial for RaiZ stability. To further define the role of RaiZ in the cell, they perform RNA-seq to identify its targets. This screen produced *hupA* as a target of RaiZ. Companion biochemical analyses are used to define the nucleotide bases involved in the sRNA-mRNA interaction. Finally, the authors examine the functional role of ProQ in this regulation, showing that ProQ assists RaiZ in blocking the ribosome from loading onto *hupA*. Notably, while RaiZ also binds the well-known chaperone Hfq and the mechanism of action of RaiZ is similar to that of traditional Hfq-binding sRNAs, the authors convincingly show that RaiZ is, in fact, acting in conjunction with ProQ.

This manuscript is of very high quality. The work is logical, the writing is clear, and the data are excellent. All of the claims made are extensively backed up by convincing data. The arguments the authors make are persuasive.

I have some minor suggestions for improvement:

1. Figure EV1 - The estimates of copy number of RaiZ/cell seem surprisingly high based on the band strength of the samples shown in the figure. Providing quantitation of these data might

strengthen the case. Can the authors check the quantification or tell us how they arrived at their conclusions?

Reply: Following the Referee's suggestion, we have quantified the signal with ImageQuant Tools and incorporated it into the figure. Please, note that we report a combined estimated copy number for both the long and the short forms of RaiZ. Based on this quantitation, we have reduced the maximal observed accumulation of RaiZ to 50-60 copies per cell.

2. Figure 2C - The authors need to specify what the difference is between the solid and dashed lines to the right of the gel. The solid and dashed lines both have the annotation "protected by ProQ".

Reply: We initially meant to distinguish between strongly and weakly protected sites. However, based on the suggestion of Referee #2, we keep only the strongly protected sites that are labelled with solid lines.

3. Figure 4C - The evidence that RaiZ-S does not affect the *phupB*-GFP fusion is difficult to interpret as the signal from the reporter is less strong than the no-GFP control. It is possible this is an artifact of the image, but if that is the case, a quantitative measurement would be better. I wonder if the labeling of the figure panels was accidentally exchanged? In both Fig EV4 and Fig 6B, the *phupB*-GFP expression is stronger than the control. Is there a labeling mistake? If not, the authors need to explain these data.

Reply: In the experiment shown in Fig 4C, all the strains express GFP (including pXG1) fused to different 5' UTRs. To ensure that the fluorescence could be accurately monitored we scaled the intensities of the strains. The *hupA*-GFP strains are translationally more efficient than the others likely because of a very strong ribosome-binding site of the *hupA* mRNA. As a result of the scaling the pXG1-containing controls and *hupB*-GFP-carrying strains appear less bright than the *hupA*-GFP strain. The plate assay is semi-quantitative, therefore we provide FACS-based quantification of fluorescence in Fig 6B, which shows that, upon normalisation for the differences in basal fluorescence of the reporters (pJV300-containing strains), overexpression of RaiZ does not repress pXG-1 or *phupB*-GFP reporters while it does decrease fluorescence of *phupA*-GFP-carrying strains. We have repeated the plate assays shown in Fig 4C and 6B several times and, indeed, found that *phupB*-GFP-containing cells are more fluorescent than the control pXG-1-containing ones, meaning that the labelling in Fig 4C was accidentally exchanged. We thank the Referee for having spotted this discrepancy and apologize for any confusion it might have created. The Fig 4C is now replaced with our newly obtained picture, which is in full agreement with data shown on Fig 6B and EV4.

2nd Editorial Decision

10 February 2017

Thank you for submitting a revised version of your manuscript. It has now been seen by two of the original referees whose comments are shown below. As you will see they both find that all criticisms have been sufficiently addressed, and I am therefore happy to inform you that your manuscript has been accepted for publication in The EMBO Journal.

REFeree REPORTS

Referee #1:

I appreciate that the authors were so conscientious in their revision. The study is an important contribution and should be published without delay.

Referee #2:

The authors have done additional experiments to verify the specific effect of ProQ on RaiZ structure. They also have taken into account all the remarks of the referees and were more careful in the interpretation of some of their data. This is really an interesting study, which shows for the first time the mechanism of action of ProQ-dependent sRNA. It fully deserves publication in EMBO J.

Corresponding Author Name: Jörg Vogel

Manuscript Number: EMBOJ-2016-96127